# NF-κB: A Druggable Target in Acute Myeloid Leukemia

**DOI:** 10.3390/cancers14143557

**Published:** 2022-07-21

**Authors:** Barbara Di Francesco, Daniela Verzella, Daria Capece, Davide Vecchiotti, Mauro Di Vito Nolfi, Irene Flati, Jessica Cornice, Monica Di Padova, Adriano Angelucci, Edoardo Alesse, Francesca Zazzeroni

**Affiliations:** Department of Biotechnological and Applied Clinical Sciences (DISCAB), University of L’Aquila, 67100 L’Aquila, Italy; barbara.difrancesco1@graduate.univaq.it (B.D.F.); daria.capece@univaq.it (D.C.); davide.vecchiotti@univaq.it (D.V.); mauro.divitonolfi@univaq.it (M.D.V.N.); irene.flati@graduate.univaq.it (I.F.); jessica.cornice@graduate.univaq.it (J.C.); monica.dipadova@univaq.it (M.D.P.); adriano.angelucci@univaq.it (A.A.); edoardo.alesse@univaq.it (E.A.); francesca.zazzeroni@univaq.it (F.Z.)

**Keywords:** NF-κB, acute myeloid leukemia, NF-κB inhibitors, cancer therapy, targeted therapy

## Abstract

**Simple Summary:**

AML is a highly heterogeneous hematological disease and is the second most common form of leukemia. Around 40% of AML patients display elevated nuclear NF-κB activity, providing a compelling rationale for targeting the NF-κB pathway in AML. Here we summarize the main drivers of the NF-κB pathway in AML pathogenesis as well as the conventional and novel therapeutic strategies targeting NF-κB to improve the survival of AML patients.

**Abstract:**

Acute Myeloid Leukemia (AML) is an aggressive hematological malignancy that relies on highly heterogeneous cytogenetic alterations. Although in the last few years new agents have been developed for AML treatment, the overall survival prospects for AML patients are still gloomy and new therapeutic options are still urgently needed. Constitutive NF-κB activation has been reported in around 40% of AML patients, where it sustains AML cell survival and chemoresistance. Given the central role of NF-κB in AML, targeting the NF-κB pathway represents an attractive strategy to treat AML. This review focuses on current knowledge of NF-κB’s roles in AML pathogenesis and summarizes the main therapeutic approaches used to treat NF-κB-driven AML.

## 1. Introduction

Acute myeloid leukemia (AML) is one of the most prevalent hematological malignancies. In fact, it is the second most common type of leukemia diagnosed in adults and children in the US, representing 31% of all adult leukemia cases [1]. AML is a genetically heterogeneous malignant disorder of the hematopoietic system arising from leukemic stem cells (LSCs). It is characterized by an uncontrolled increase in the number of immature myeloid cells in the bone marrow leading to hematopoietic insufficiency, with or without leukocytosis [2,3].

AML is now curable in approximately 35–40% of patients younger than 60 years of age, while for people >60 years of age the prognosis is certainly worse, with a cure rate of <20% [4,5]. To date, conventional approaches (i.e., standard “7 + 3” chemotherapy and allogenic stem cell transplantation) [5] are still the clinical gold standard. Moreover, several innovative agents are now successfully used for treating AML patients, such as CAR-T cell therapy, although also in this case important hurdles have to be faced including high costs and the absence of predictive markers [6,7]. Despite the considerable progress in the treatment of AML over the past decades, recurrent disease is still a major clinical problem, with a median survival of 6 months [6]. Moreover, the survival of patients is frequently unpredictable depending on many factors including age, the overall health of the patient and the oncogenic genetic/molecular background defining the tumor subtype [4]. Accordingly, the state of art suggests that a significant improvement in treatment decisions could arise from the evolution of the molecular classification of AML subtypes, overcoming the current limitation of morphological/cytogenetic-based classification.

NF-κB is a regulator of many biological functions such as cell survival, apoptosis, invasion and hematopoiesis [8,9], where it regulates hematopoietic stem cells’ (HSCs) self-renewal and differentiation in myeloid and lymphoid lineage [10]. The family of NF-κB transcription factors includes five proteins known as RelA/p65, RelB, c-Rel, NF-κB1 (p105/p50) and NF-κB2 (p100/p52) that homodimerize or heterodimerize to form different NF-κB complexes. The heterodimer p65/p50 is the most abundant active NF-κB complex in mammalian cells [8,11,12,13]. Under physiological condition, NF-κB is in an inactive state in the cytoplasm through its interaction with NF-κB inhibitory proteins (IκBs) [8,11,12,13]. The activation of NF-κB happens via two signaling pathways known as the classical (canonical) and the alternative (non-canonical) pathway. The canonical NF-κB pathway is activated by several stimuli such as proinflammatory cytokines, lipopolysaccharide (LPS) and growth factors, that in turn activate the IκB kinase (IKK) complex. Next, IKK complex promotes IκB phosphorylation, thus triggering IκB poly-ubiquitination and proteasomal degradation. Activated NF-κB translocates to the nucleus, where it activates the transcription of its target genes. In the non-canonical pathway, the activation of p100/RelB complex relies on NF-κB inducing kinase (NIK), that phosphorylates and activates IKKα complex, leading to the degradation of p100 to p52 and subsequent nuclear translocation of the p52/RelB complex [11,14].

The constitutive activation of NF-κB is a hallmark of most human tumors, including AML, where it promotes cell survival and resistance to therapy by activating anti-apoptotic genes and promoting inflammation in the tumor microenvironment (TME) [15,16]. In AML, NF-κB is frequently stably activated by recurrent genetic alterations of upstream regulators of its pathway [17,18,19]. However, constitutive NF-κB activation can also ensue from inflammatory stimuli and other cues emanating from the TME or oncogenic alterations lying outside the traditional NF-κB pathway, as discussed below [8].

This review focuses on the role of NF-κB in the pathogenesis of AML and provides an overview of the current therapeutic strategies targeting the NF-κB pathway in AML patients. This may support the exploration of new potential targets to overcome the side effects due to the dose-limiting toxicities of a systemic NF-κB blockade.

## 2. NF-κB Pathway in AML Pathogenesis

NF-κB signaling has a central role in AML carcinogenesis, where it is responsible for differentiation, survival, growth of leukemia cells and resistance to therapy [17,18,19]. Indeed, about 40% of patients with AML exhibited increased activity of NF-κB [20]. In particular, it has been demonstrated that NF-κB is constitutively active in CD34^+^ stem cells from M3, M4 and M5 AML patients [21]. In AML, NF-κB, constitutive activation seems to be crucial for maintaining AML cells (i.e., proliferation and survival) rather than promoting myeloid transformation [18]. In addition, aberrant NF-κB signaling is correlated with resistance of AML cells to radiation and chemotherapy [18]. Evidence has shown that NF-κB mediates chemoresistance in AML, based on its ability to induce anti-apoptotic genes such as *BCL-2* (*B-cell lymphoma 2*) and *BCL-X_L_* (*B-cell lymphoma-extra-large*) [22]. Recently, Wei and collaborators demonstrated that Aurora A-dependent NF-κB signaling drives chemoresistance in AML and was associated with worse clinical outcomes. In particular, the authors showed that TRAF-interacting protein with FHA domain (TIFA), which interacts with Tumor necrosis factor receptor (TNFR)-associated factor 6 (TRAF6) to activate the IKK complex, mediated Aurora A-dependent NF-κB activation. In keeping with this observation, TIFA silencing increased chemotoxicities of both AML cells and patient-derived peripheral blood mononuclear cell (PBMC) via inhibition of inflammatory cytokines secretion, leading to tumor necrosis factor α (TNFα)-dependent NF-κB survival pathway attenuation and, thus identifying TIFA as a potential target in AML [23]. Notarbartolo and colleagues demonstrated that NF-κB is important for the establishment of drug resistance in AML by controlling the P-gp (P-glycoprotein)-mediated expression of *MDR1* (*multidrug resistance mutation 1*) gene [24]. Several studies showed that overexpression of heterodimer p50/p65 in resistant variant AML cell lines induced *P-gp* gene and *IAP-family* genes expression, underlying the importance of NF-κB proteins in promoting chemoresistance, tumor progression and a poor prognosis [25,26,27].

NF-κB plays a pivotal role also in LSCs, where it regulates survival, proliferation and chemoresistance [28]. Zhou and collaborators demonstrated that NF-κB fosters stem-like properties (i.e., self-renewal capacity) of AML cells via LIN28B activation. Accordingly, NF-κB inhibition reduces *LIN28B* expression and cell survival as well as LSCs’ self-renewal in vitro, suggesting that inhibition of NF-κB could be a potential opportunity to kill AML cells and LSCs in order to counteract cancer resistance and disease relapse [29].

## 3. Genetic Alterations Drive NF-κB Constitutive Activation in AML

Chromosomal aberrations involving *c-Rel*, *RelA*, *NF-**κB1* and *NF-**κB2* genes were found in many hematopoietic and solid tumors. However, in AML, NF-κB constitutive activation ensues from mutations affecting genes involved in the control of NF-κB activity, as detailed below (Figure 1).

### 3.1. ATM

Grosjean-Raillard and collaborators demonstrated that activated serine/threonine protein kinase ATM (Ataxia Telangiectasia Mutated), a principal regulator of cell cycle checkpoints in response to DNA damage, triggered IκB kinase NEMO (NF-κB essential modulor) activation and p53-induced death domain protein (PIDD), resulting in NF-κB activation. Pharmacological inhibition of ATM was shown to prevent its autophosphorylation and its interaction with NEMO, leading to redistribution of NEMO and NF-κB to the cytoplasm and apoptosis of malignant myeloblasts [30,31]. These results demonstrate that constitutive phosphorylation of ATM is crucial for NF-κB activation in AML. In addition, the authors demonstrated that constitutive activation of Fms related receptor tyrosine kinase 3 (FLT3) signaling resulted in activation of NF-κB via IKK. The inhibition of FLT3 reduced NF-κB activity and promoted apoptosis in AML cell lines and CD34^+^ primary AML cells [32]. Notably, NEMO-IKK complex promoted the activation of canonical NF-κB pathway in AML [21], thus suggesting that the use of NEMO-binding domain peptides could represent an alternative strategy to indirectly inhibit NF-κB in this cancer.

### 3.2. BCR::ABL

BCR::ABL translocation, typically found in chronic myeloid leukemia (CML), has been observed also in AML patients. In fact, BCR::ABL^+^ AML, now classified as a high risk AML, is a rare subtype of AML (0.3–2% of cases) and the prognosis depends on the cytogenetic/molecular landscape [33,34,35]. Studies conducted by different research groups demonstrated that BCR::ABL translocation contributed to IKK-dependent constitutive activation of NF-κB. Accordingly, genetic or pharmacologic inhibition of NF-κB induced cell death in BCR::ABL^+^ cells [36,37]. The authors hypothesized that BCR::ABL fusion protein may activate protein kinase D2 (PRKD2), which in turn induces IKK2-dependent phosphorylation and degradation of IκBα, leading to NF-κB activation [36,37].

### 3.3. RUNX1

Another transcription factor responsible for aberrant activation of NF-κB in AML is Runt-related transcription factor 1 (RUNX1), also known as AML1. In fact, chromosomal alterations as well as point mutations affecting *AML1/RUNX1* gene have been observed in human leukemia, and patients with AML1 mutations exhibited poor clinical outcomes, underlying the important role of RUNX1 during hematopoiesis [38,39,40]. Studies conducted by Nakagawa and collaborators demonstrated that AML1/RUNX1 prevented NF-κB pathway activation in hematopoietic cells by inhibiting IKK activity. Accordingly, AML1 was shown to induce the expression of mir-223, which restrains IKK expression [41,42,43]. In keeping with these observations, mutated AML1/RUNX1 (AML1::ETO fusion product) failed to attenuate IKK kinase activity, leading to aberrant NF-κB activation. Furthermore, pharmacological inhibition of NF-κB signaling reduced tumorigenesis in vivo by suppressing NF-κB-mediated excessive proliferation of AML1/RUNX1 mutated leukemia cells [44].

### 3.4. C/EBPα

Another important bZIP transcription factor involved in myeloid development and mutated in 10–15% of AML patients is CCAAT/enhancer-binding protein alpha (C/EBPα) [45,46,47]. The presence of double mutations of C/EBPα in AML patients has been associated with a favorable prognosis [48]. Studies conducted by Paz-Priel et al. demonstrated that C/EBPα and its mutated forms interacted with the p50 subunit of NF-κB and induced the espression of NF-κB target genes (i.e., *BCL-2* (*B-cell lymphoma 2*) and *c-FLIP* (*FLICE-like inhibitory protein*)) in vitro. In turn, the p50 subunit controlled C/EBPα expression, thus generating a positive feedback loop. The authors also showed that these proteins, in complex with p50, can directly regulate the expression of the anti-apoptotic genes *BCL-2* and *c-FLIP* [45,49,50]. Moreover, Pulikkan and collaborators showed that C/EBP transcription factors also regulated IKK expression by inducing the expression of mir-223. Accordingly, repression of mir-223 due to C/EBPα mutation increased IKK-NF-κB activity [51].

### 3.5. q Deletion

Among genetic mutations, chromosome 5q deletions (del(5q)) are commonly found in high risk myelodysplastic syndromes (MDS)/AML and are associated with favorable prognosis if the percentage of blasts in the bone marrow is less than 5% [52,53]. The main actor implicated in del(5q) is miR-146a, which, under normal conditions, inhibits TRAF6. In vivo studies showed that miR-146a loss in mouse hematopoietic stem and progenitor cells (HSPC) promoted MDS/AML by increasing TRAF6 expression and NF-κB activation [54,55,56]. Pharmacological NF-κB inhibition or genetic inhibition of TRAF6 in AML cells induced G2/M arrest and apoptosis, suggesting that inhibition of TRAF6/NF-κB axis could be a potential approach to treat AML patients with del(5q) and low miR-146a levels. The same results were obtained in miR-146a-depleted HSPC [53]. Further analysis clarified that NF-κB-mediated leukemic cell survival was mediated by sequestosome 1 (SQSTM1, also known as the ubiquitin-binding protein p62), a scaffold protein of the TRAF6/NF-κB axis, whose overexpression is induced by NF-κB, generating a positive feedback loop. Knockdown of p62 resulted in reduced colony formation or tumor growth and reduced TRAF6 activation and NF-κB nuclear localization both in vitro and in vivo, thus indicating that p62 sustains NF-κB activation and leukemic cell functions (i.e., cell cycle and myeloid cell development) through TRAF6. Overall, these findings support the hypothesis that theh TRAF6/p62/NF-κB axis is responsible for leukemic cell survival in del(5q) AML with low miR-146a [53].

### 3.6. RAS

It is well demonstrated that NF-κB is also activated by N-RAS and K-RAS mutations that occur in approximately 20% of AML cases [57,58,59]. Birkenkamp and collaborators showed an important association between constitutive NF-κB activity and persistent Ras/phosphoinositide 3-kinase (PI3K)/protein kinase B (PKB) signaling in human AML blasts, where these two different signaling pathways sustain the survival of AML cells. Althought activating mutations in FLT3 and c-Kit (receptor tyrosine kinase (RTK)), which encode receptor tyrosine kinases upstream of Ras, were shown to be responsible for aberrant activation of Ras signaling in 40% of AML cases [59], Birken and collaborators demonstrated that NF-κB activation in AML blasts was not dependent on FLT3 mutations, but was triggered by RAS and PI3K/AKT (protein kinase B (PKA)) pathways, as the pharmacological inhibition of these two signaling blocked NF-κB activity [60]. Nevertheless, the role of FLT3 remains controversial, as some reports in the literature showed that FLT3 overexpression, and/or mutations (i.e., FLT3-ITD (internal tandem duplication), the most common genetic abnormality), can induce canonical or non-canonical NF-κB signaling in AML. Accordingly, genetic or pharmacological inhibition of FLT3 reduces NF-κB activity and promotes apoptosis of both AML cell lines and primary cells [32,61,62] (Figure 1).

## 4. Pro-Inflammatory Microenvironment and NF-κB Activation in Leukemic Cells

Beyond genetic alteration, aberrant NF-κB activation could also stem from the steady exposure of tumor cells to inflammatory stimuli and other cues emanating from the TME.

Inflammation-induced tumorigenesis is one of the most important mechanisms underpinning the proliferation of leukemic cells. It is known that NF-κB is the major player between inflammation and cancer [63,64] and contributes to evading apoptosis and sustaining cell survival by inducing the secretion of pro-inflammatory cytokines (i.e., TNFα, Interleukin 6 (IL-6), Interleukin 1β (IL-1β) and regulating heme oxygenase-1 (HO-1) expression and inducible nitric oxide synthase (iNOS) activation in the TME [65,66,67]. Although the increased activation of iNOS is associated with pro-apoptotic functions, the presence of high iNOS expression in AML patients suggests that it could either promote or inhibit apoptosis during carcinogenesis, probably depending on inflammation status [68,69]. As reported in the literature, a subset of AMLs is characterized by secretion of high levels of pro-inflammatory cytokines, that in turn activate NF-κB pathway, thus generating a positive feedback loop able to sustain NF-κB-dependent AML growth both in patients and in murine models [18]. The role of TME in fostering tumor progression by paracrine secretion of cytokines is well established [15,70]. Jacamo and collaborators demonstrated that bone marrow stromal cells promoted NF-κB activation in AML cells through vascular cell adhesion molecule 1 (VCAM-1) and very late antigen 4 (VLA-4) interaction [71] and highlighted the importance of surrounding stroma in induction and maintenance of aberrant NF-κB activation in tumor cells [9,72,73]. Recent findings showed that the immune modulator Interferon regulatory factor 2 binding protein 2 (IRF2BP2) attenuates the inflammatory signals between monocytic AML cells by controlling the NF-κB-mediated TNFα signaling. Accordingly, the suppression of IRF2BP2 induces caspase 8- and caspase 3-mediated apoptotic cell death via NF-κB mediated upregulation of IL-β1 [74,75]. In keeping with these findings, Volk and collaborators demonstrated that TNF stimulated leukemia cell growth via autocrine induction of NF-κB and JNK-AP1 (c-Jun N-terminal kinases/activator protein-1) signaling, suggesting that several signaling pathways cooperate to sustain cancer cell proliferation. In fact, solely NF-κB inhibition was not sufficient in inhibiting AML tumor growth, due to the activation of anti-apoptotic genes by the TNF/JNK axis. Accordingly, in vivo studies demonstrated that the inhibition of NF-κB along with TNF inactivation induced leukemic cell death. Additionally, the authors showed that TNF produced by leukemic cells inhibited normal HSPC growth in a paracrine manner, thus finding a possible explanation for the hematopoietic repression observed in AML patients. These findings suggest that co-inhibition of both TNF/JNK-AP1 and TNF/NF-κB pathways could be a potential therapeutic approach to inhibit leukemic cell growth while protecting HSPC in those AML Fab subtypes which express TNF, such as M3, M4, M5 [76]. An additional study underscoring the paramount role of TNF/NF-κB axis in AML was published by Kagoya and collaborators. The authors demonstrated that LSCs, also known as leukemia-initiating cells (LICs), showed constitutive NF-κB activity due to autocrine TNFα secretion, that in turn promoted IκBα degradation through prolonged activation of proteasome machinery, induction of IκBα degradation and translocation of NF-κB into the nucleus. Since TNFα is one of the NF-κB target genes, leukemia progression is maintained by the NF-κB/TNFα feedback loop [77,78]. Conversely, NF-κB inhibition in LICs curbs tumorigenesis in vivo, suggesting that the NF-κB/TNFα axis supports leukemia progression [78]. Furthermore, Li and collaborators demonstrated that treatment with TNF and IL-1β inhibitors sensitized LSCs to NF-κB inhibition indicating that this combination strategy could be used to remove leukemic cells as well as LSCs and overcome the LSCs-mediated drug resistance [79] (Figure 1).

## 5. Aberrant NF-κB Activation by NF-κB Regulators/Interactors

The hyperactivation of NF-κB in leukemic cells is also due to an increased activation of upstream regulators of the NF-κB pathway [80,81], making these components promising targets for cancer treatment. Several studies identified Interleukin-1 receptor accessory protein (IL-1RAP), Interleukin-1 receptor-associated kinase 1 (IRAK1), transforming growth factor-β activated kinase (TAK1) and Bruton’s tyrosine kinase (BTK) as overexpressed in primary AML cells. Accordingly, inhibition of these proteins suppressed NF-κB activation and restrained tumor growth by promoting apoptosis both in vitro and in vivo [18,82,83,84,85].

The proviral insertion in murine (PIM) lymphoma proteins are proto-oncogenic serine/threonine kinases, constitutively active in AML [86,87]. Recent studies demonstrated that PIM2 supports AML tumorigenesis by suppressing apoptosis and inducing cancer cell survival via NF-κB activation [88]. Indeed, PIM2 activates the NF-κB pathway by inducing the phosphorylation of serine threonine kinase Cot and consequently the upregulation and degradation of IκB. Furthermore, Nihira and collaborators demonstrated that Pim1 controls the NF-κB pathway by phosphorylating RelA/p65 at Ser276 [89]. On the other hand, it has been demonstrated that NF-κB regulates the expression of Pim-1 and 2 kinases through CD40 signaling and in response to FLT3/ITB oncogenic mutants, respectively [88,90].

Other important players promoting constitutive activation of the NF-κB pathway are proteins involved in NF-κB-mediated gene transcription such as Bromodomain-containing protein 4 (BRD4) and murine double minute 2 (MDM2). BRD4 belongs to bromodomain and extra terminal domain (BET) protein family and interacts with acetylated lysine in histone or non-histone proteins [91,92]. BRD4 plays a central role in transcriptional regulation via interactions with specific proteins like positive transcription elongation factor B (pTEFb). With regard to NF-κB pathway, although the molecular mechanism is still poorly understood, BRD4 might bind to acetylated histones and proteins, such as p65, leading to transcription of the NF-κB-regulated pro-inflammatory genes [92,93,94,95]. Silencing of BRD4 reduced leukemic tumor growth in mice [96]. Moreover, pharmacological inhibition of BRD4 with BEF inhibitors restrained NF-κB-mediated inflammatory response [97]. Accordingly, treatment with I-BET762, a pan-BET inhibitor, suppressed inflammation and reduced the incidence of death in mice after prolonged lipopolysaccharide (LPS) exposure [98].

It has been demonstrated that MDM2 induces p65 expression in different cells; MDM2 overexpression was observed in AML, but its capacity to interact with NF-κB remains debated [99,100]. On the other hand, overexpression of negative regulator MDM2 inactivates p53, which is frequently inactivated in AML cell lines and patients and whose loss is associated with poor prognosis. Recently, it has been demonstrated that inhibition of the p53-MDM2 complex promoted antitumor activity in vivo [101].

Constitutive NF-κB expression is also associated with the enhanced activity of an immunoproteasome variant expressed in hematopoietic cells, functionally close to “classical” proteasome, which represents an attractive target in hematologic malignancies [102]. Accordingly, increased activation of this proteasome variant has been observed in CD34^+^ leukemic cells and in LICs from different murine models [78,103,104,105] (Figure 1).

## 6. Alternative NF-κB Pathway in AML

Suppression of the canonical NF-κB pathway reduces AML growth [36], supporting the use of NF-κB inhibitors as potential drugs in AML, either alone or in combination with current treatment. A limited number of studies highlighted the opposite roles of the non-canonical NF-κB pathway in cancer. In AML, activation of the alternative NF-κB signaling pathway (i.e., upregulation of NF-κB inducing kinase (NIK) and p52) was shown to promote cell differentiation. In contrast, a study conducted by Shanmugan demonstrated that p52 repressed *DAPK1* transcription that in turn contributed to increased apoptosis of AML cells [62]. In keeping with the anti-tumoral activity of the non-canonical NF-κB pathway, Xiu and collaborators demonstrated that NIK stabilization suppressed (mixed lineage leukemia) MLL-AF9-induced AML in vivo via activation and inhibition of alterative and canonical NF-κB pathways, respectively. They also showed that NIK stabilization induced DNA (cytosine-5)-methyltransferase 3A (DNMT3A) and Notch upregulation, as well as downregulation of Myocyte Enhancer Factor 2C (MEF2C) and RelA, suggesting that NIK stabilization could be an attractive therapeutic strategy to treat specific AML tumors. However, further studies are needed in order to clarify the controversial role of NF-κB alternative pathways in AML [106]. Accordingly, treatment with Verteporfin, an FDA-approved photosensitizer, reduced AML development in vivo, through upregulation of RelB and p52, members of NF-κB alternative pathway, mimicking the effects of NIK stabilization [106] (Figure 2). Recently, Zhao and collaborators demonstrated that *TRAF-interacting protein with forkhead-associated domain B* (*TIFAB*), a direct target gene of RelB overexpressed in human AML and correlated with poor prognosis, upregulates HOXA9, thus promoting MLL-AF9 AML in vivo. In addition, the authors showed that TIFAB controls the maintenance and proliferation of MLL-AF9 LSCs through the upregulation of several genes including *Six1*, *Msi2*, *Flt3 and Hoxa cluster genes* [107].

## 7. Targeting NF-κB Pathway in AML

In recent years, the identification of specific gene mutations and their aberrant protein products opened the door towards individualized approaches for treating AML with new specific targeted drugs. Due to the importance of NF-κB in AML tumorigenesis, the inhibition of the NF-κB pathway represents a promising target for treating refractory AML patients, as well as for improving their prognosis. To date, different approaches have been proposed to achieve the best efficacy in targeting NF-κB pathway in AML (Figure 3).

## 8. NF-κB Core Pathway Inhibitors

### 8.1. Proteasome Inhibitors

The ubiquitin-proteasome system is a complex fundamental protease that degrades ubiquitinated proteins involved in the regulation of numerous cellular processes, such as cell cycle progression, cell growth and proliferation, DNA repair, immune and inflammatory response, processing of antigens presented in association with the major class I histocompability complex (MHC I) and degradation of mutated or damaged proteins. NF-κB activity is under the control of the ubiquitin-proteasome system, as it is responsible of ubiquitination and consequent proteolysis of IκBα, the NF-κB inhibitor, which sequesters NF-κB in the cytoplasm when inactive [108].

The most important drugs that inhibit NF-κB activation and are also effective for AML treatment are proteasome inhibitors.

Bortezomib was the first proteasome inhibitor approved for clinical use [109,110]. When Bortezomib was tested in AML patients in combination with conventional drugs, it did not improve their outcomes [82,111,112,113,114,115,116]. However additional clinical trials of Bortezomib in combination with chemotherapeutic agents are still ongoing (NCT01371981; NCT00510939; NCT01861314; NCT01420926). Bortezomib acts by binding reversibly the b5-subunit of the proteasome, thus blocking the chymotrypsin-like activity and the degradation of IκBα, thus triggering apoptosis. Liu and colleagues showed that, in AML, Bortezomib blocked the binding of Sp1 (Sp1 transcription Factor)/NF-κB complex to the DNA methyl-transferase 1 (DNMT1) gene promoter, leading to a reduced DNA methyltransferase activity that, in turn, induced DNA hypomethylation and silenced gene transcription. The same group demonstrated that bortezomib arrested tumor growth by blocking Sp1/NF-κB/Histone deacetylase (HDAC) complex, upregulating miR-29b downregulating *c-KIT* gene and, finally, inibiting aberrant TK activity both in vitro and in vivo [117,118]. Notably, Bosman and collaborators showed that bortezomib failed to achieve a strong inhibition of NF-κB activity in AML stem cells, that are, therefore, insensitive to this drug [119]. A first clinical trial conducted in pediatric patients with resistant AML showed little activity of bortezomib as a single agent [120]. Given the poor effectiveness of bortezomib as a single agent, synergistic effects between bortezomib and cytotoxic therapy have been evaluated. Evidence showed that bortezomib synergises with other chemotherapeutics (i.e., doxorubicin, daunorubicin) by intensifying the inhibition of the NF-κB pathway and p53 activation [121]. When administrated with histone deacetylase inhibitors, bortezomib disrupts NF-κB activity, inhibits AKT signaling and downregulates anti-apoptotic proteins [77]. In addition, to trigger apoptosis in leukemic cells, bortezomib reduced the frequency and function of LSCs and extended the overall survival of MLL-rearranged AML in vivo. The authors demonstrated that bortezomib suppressed the self-renewal ability of LSCs by inhibiting the recruitment of NF-κB to Cyclin dependent kinase 6 (CDK6) promoter and consequently the expression of CDK6. Since bortezomib acts selectively against LSCs with little side effects againts normal HSPC, these studies supports the potential use of bortezomib in AML patients with MLL rearrangements [122].

In the last decade, a second-generation of proteasome inhibitors has been developed, including the immunoproteasome inhibitors carfilzomib, in order to overcome side effects observed after treatment with bortezomib. In vitro studies conducted by different research groups demonstrated that carfilzomib induced apoptosis of primitive AML blasts and a reduction of quiescent CD34^+^/CD38^−^ cells. However, the clinical effectiveness of these drugs in AML is still under investigation [18].

The proteasome inhibitor MG-132 (Carbobenzoxy-l-leucyl-l-leucyl-l-leucinal) belongs to the class of peptide aldehydes and induces apoptosis of leukemic cells, including CD34^+^/CD38^−^ LSCs, by increasing phosphorylated IκBα and reducing the expression of NF-κB target genes. Notably, MG-132 does not affect normal HSCs, which do not rely on NF-κB for survival [20,123,124]. However, due to its low specificity and low bioavailability, MG-132 is not present in the pipeline for AML treatment [77].

### 8.2. IKK Inhibitors

Another group of NF-κB inhibitory compounds are represented by small-molecule IKK inhibitors [125]. IKKβ inhibitor AS602868 was shown to induce cell death in human primary AML cells both in vitro and in vivo, without affecting normal HSCs. The apoptotic response observed in these cells following AS602868 treatment was due to mitochondrial transmembrane potential disruption with consequent activation of the caspase cascade [126]. Jordhein and collaborators demonstrated that AS602868 triggered apoptosis and influenced immune-related genes expression in fresh human AML blasts. They also assessed the sensitivity of AML blasts to AS602868 in combination with other chemotherapeutic drugs such as AraC, etoposide and daunorubicin, suggesting that combination therapy seems to be more effective than single AS602868 therapy [127].

BMS-345541 is another drug that interacts and inhibits IKK, leading to decreased IκB phosphorylation and thereby NF-κB activation in several cancers [128,129]. A study conducted by Reikvam showed that NF-κB inhibition by BMS-345541 changed the genetic expression profile in primary AML cells. Specifically, bioinformatical analyses displayed alteration in cytokine/interleukin signaling, metabolic systems, transcriptional regulation and immune system/cellular comunication, all events important in AML. Moreover, the authors observed changes in transcriptional regulation of several genes including *RUNX1*, *CEBPα*, *NFκB1*, *BID* (*BH3-interacting domain death agonist*), *CXCL10* (*C-X-C motif chemokine ligand 10*) and *IL-1* (*Interleukin-1*), as well as genes associated with LSCs, suggesting that inhibition of NF-κB significantly remodels the expression of genes important for leukemogenesis and could be a possible approach to target LSCs [130].

In vitro studies led by Harikumar and colleagues showed that the antioxidant flavonoid xanthohumol, a prenylated chalcone, promoted TNF-induced apoptosis in AML cells. Significantly, the observed cell death was associated with downregulation of NF-κB-regulated anti-apoptotic genes. NF-κB blockade was achieved via inhibition of both IKK activation and binding of p65 to DNA [131].

The sesquiterpene lactone parthenolide (PTL) is a natural compound that acts as a NF-κB inhibitor by binding IKK and modifying the heterodimer p50/p65 [132]. Studies conducted by Guzman and colleagues demonstrated that PTL induced apoptosis in primary human AML cells, progenitor and LSCs by inhibiting the NF-κB pathway, activating p53, increasing reactive oxygen species (ROS) and JNK activation both in vitro and in vivo [132,133]. However, the poor bioavailability makes PTL ineffective as a drug. A new PTL analog, dimethyllamino-parthenolide (DMAPT), exhibited a cytotoxic effect on primary human LSCs and on expansion of the leukemic cell population. No effects were observed on normal cells [134]. As for PLT, DMAPT blocks the NF-κB signaling pathway, induces oxidative stress response and activates the p53 pathway both in vivo and in vitro [134]. Recently, Darwish and collaborators designed and tested in AML cells poly lactide co-glycolide (PLGA) nanoparticles conjugated with anti-CD44 and encapsulating PTL (PLGA-antiCD44-PTL-NPs). They demonstrated that PTL was succesfully uptaken in CD44-overexpressed AML cells compared to normal cells and observed also in tumor cells a significant reduction of cell proliferation [135]. The same results were obtained encapsulating antitumor drugs (i.e., cisplatin, paclitaxel, doxorubicin) in PLGA [136].

As seen for PTL and DMAPT, melampomagnolide B (MMB) showed anti-tumoral activity due to its ability to inhibit the NF-κB pathway by inhibiting IKKβ activation in several cancer cell lines [137,138]. Accordingly, MMB also promotes apoptosis by increasing ROS levels and inhibiting the mitochondrial glutathione system, thus leading to mitochondrial dysfunction in cancer cells [139]. Janganati and collaborators demonstrated that MMB triazole analog 7 h, an inhibitor of p65 phosphorylation, exhibited anti-cancer effects in many tumor cell lines, including AML cell lines. Specifically, 7 h was shown to interact with and block IKKβ activation, leading to inhibition of p65 phosphorylation in primary AML cells [140].

Another potent inhibitor of IKK/NF-κB signaling is celastrol, a quinone methide triterpenoid, able to suppress NF-κB-regulated genes expression induced by various stimuli including TNFα, LPS, as well as the DNA-binding of NF-κB in different cell lines. Celastrol inhibits IKK activity and exhibits anti-tumoral activity both in vitro and in vivo [141]. Recently it has been demonstrated that celastrol synergized with embelin, a X-linked inhibitor of apoptosis protein (XIAP) inhibitor, to promote antiproliferative and pro-apoptotic effects in AML cell lines [142].

BAY 11-7082 is a poorly selective IKK inhibitor that exerts its anti-cancer and anti-inflammatory effects in several cancer cell lines by targeting multiple systems [143,144]. As for celastrol, BAY 11-7082 was shown to reduce p65 and pp65 levels, thus inducing anti-proliferative and antiangiogenic effects in the OCI-AML3 cell line [145].

Curcumin (1,7-bis(4-hydroxy-3-methoxyphenyl)-1,6-heptadiene-3,5-dione) is a natural polyphenol with antioxidant, anti-inflammatory, antimutagenic, antimicrobial and anticancer properties [146,147]. Several studies demonstrated that curcumin targets multiple signaling pathways, including NF-κB. While as an anti-inflammatory agent this polyphenol counteracts NF-κB activation and suppresses inflammation via different mechanisms [146], as an anticancer drug curcumin exhibits tumor suppressive function by inhibiting proliferation, drug resistance and promoting apoptosis through inhibition of IKK-mediated NF-κB activation and other factors (i.e., AP-1, MAPK, AKT, COX-2, P-gp) [17]. The antitumor effects of curcumin were observed also in HL-60R cells (multidrug resistant HL-60 cells), where it was shown to block P-gp function, probably due to NF-κB inhibition. Several other studies demonstrated that isoxazole, one of the numerous derivatives of curcumin, was able to suppress HL-60 cell growth and promote cell death, in vitro. Despite the very promising evidence in vitro, the clinical use of curcumin and its derivates remains limited [17].

Heat shock protein 90 (HSP90) is a chaperone protein, overexpressed in cancer, essential for IKKα stabilization and consequently for NF-κB signaling [148]. Several HSP90 inhibitors have been developed and showed anti-tumor effects in cell lines and AML patients (i.e., NVP-AUY922-AG, SNX-5422). Most of them were tolerated in advanced AML patients in phase I clinical trials [149,150,151]. A study conducted by Qin and colleagues demonstrated that SNX-2112, belonging to the second generation of HSP90 inhibitors, inhibited cell proliferation and promoted differentiation, G2/M cell cycle arrest and apoptotic cell death in human AML cells via AKT and NF-κB inhibition. In particular, the authors showed that SNX-2112 downregulated IKKα, IKKβ and Akt expression in AML cells leading to IκB upregulation, NF-κB/p65 inhibition and modulation of PU.1 and C/EBPα expression [151]. Accordingly, another HSP90 inhibitor, NVP-AUY922-AG, induced AML apoptosis by PI3K and IKK inhibition and increased HSP70 expression in both AML cell lines and primary cells. Moreover, NVP-AUY922-AG was shown to synergize with AraC when used as combination therapy [149].

## 9. Indirect NF-κB Inhibitors

In light of numerous on-target toxicities observed following the inhibition of core components of NF-κB pathways, mainly caused by systemic blockade of relevant physiological functions excerted by NF-κB, novel drugs, targeting nonredundant, upstream activators or downstream effectors as well as interactors of NF-κB pathway (i.e., BTK, TK, c-IAP1, XIAP, BCL-2 and Growth arrest and DNA damage-inducible 45 B (GADD45B)) have been investigated [109].

## 10. Inhibitors of Upstream Regulators of NF-κB

### 10.1. Tyrosine Kinases Inhibitors (TKI)

It is well known that in AML, the NF-κB pathway is activated by overexpression or activating mutations of FLT3 [61]. Several FLT3 inhibitors have been developed during the last few years in order to test their clinical benefit, as well as their specificity both as monotherapy and combination therapy [18,152,153,154]. Birkenkamp et al. demonstrated that NF-κB pathways were not affected by FLT3 inhibition in primary AML blasts, suggesting that redundant activation pathways are present in this cell model [60]. Thus, additional studies are needed to understand the potential effectiveness of FLT3 inhibitors as monotherapy to induce cytotoxicity via NF-κB repression. In contrast, the combination therapy between FLT3 and NF-κB inhibitors showed promising results both in vitro and in vivo. As reported by Griessinger, the combination of FLT3 and IKK2 inhibitors triggered apoptosis and induced cell proliferation arrest of the human FLT3 dependent-AML cell line [155]. Accordingly, Wang and collaborators reported the antitumor effect of the FLT3 selective inhibitor, SC203048, combined with the NF-κB inhibitor PTL, in an AML xenograft model. Following combination therapy, the authors observed reduced tumor growth and increased apoptosis along with decreased expression of wild-type FLT3, p65, and NF-κB regulated genes such as *cyclin D1* and *BCL-2*, suggesting the potential use of these inhibitors for treating AML [156].

Midostaurin was the first TKI FDA-approved for treatment of newly diagnosed FLT3-mutated AML in combination therapy with chemotherapy. Midostaurin is a potent multikinase inhibitor that acts on FLT3, c-KIT, Platelet-derived growth factor receptor (PDGFRs), Vascular endothelial growth factor Receptor (VEGFR) and protein kinase C [157]. Several studies showed the prolonged overall survival (OS) of AML patients with FLT3 mutation when treated with Midostaurin in combination therapy (i.e., CDDO-Me, rapamycin, decitabine, UO126) rather than as monotherapy [158,159]. Similar results were obtained when a combination of Midostaurin with Vyxyeos (cytarabine + daunorubicin) was tested [160].

Another important drug class that could be potentially used for treating AML by indirectly suppressing the NF-κB pathway is represented by Bruton’s Tyrosine Kinase inhibitors (BTKs). BTK is an important regulatory member of B cell receptor (BCR) signaling, and the most representative agent targeting BTK is ibrutinib. BTK-activating mutations occurs in 50% of AML patients and constitutive activation of BTK was observed in AML [161,162,163]. Rushworth and collaborators demonstrated a decrease in NF-κB, ERK (extracellular signal-regulated kinase) and AKT activity in primary AML and AML cell lines following treatment with ibrutinib, leading to cell death and suggesting that BTK supports AML cell survival [163]. In addition, to block proliferation via inhibition of cytokines and chemokines expression in vitro, the authors showed that ibrutinib also inhibited AML cell crosstalk with bone marrow stromal cells (BMSCs), thus disrupting the interplay between tumor cells and TME and increasing the efficacy of conventional chemotherapies. In contrast, the authors showed no effectiveness of ibrutinib in treating AML cells in which the survival is supported by TNFα, suggesting that this drug could be used only in AML cells expressing low levels of TNFα [163]. Considering the good response demonstrated by other hematological malignancies to BTK inhibitors, the use of ibrutinib in AML patients remains an active field of study.

### 10.2. Serine-Threonine Kinases Inhibitors

High levels of TGFβ-activated kinase 1 (TAK1) in AML CD34^+^ cells compared to normal cells suggest that the pro-survival functions of NF-κB are also mediated by TAK1 [84]. Pharmacological inhibition (AZ-TAK1, 5z-7-oxozeaenol) and genetic ablation of TAK1 promoted NF-κB downregulation, leading to cell death both in vitro and in vivo. Specifically, Bosman and colleagues showed that TAK1 inhibition reduced p-IκBα levels and p65 activity. Importantly, overexpression of NF-κB after TAK1 inhibition, rescued cancer cells from apoptosis [84].

Overexpression of IRAK1 in AML patients with or without del(5q) is mediated by miR-146a and Toll-like receptor family (TLR)1/2/6, respectively. IRAK1 inhibitors induced a dose-dependent reduction of cell growth and induction of apoptosis in BCL-2-independent AML cells. Pacritinib was developed as an oral inhibitor of Janus Kinase 2 (JAK2), and successive analyses demonstrated that it also inhibits IRAK1. Hosseini and collaborators demonstrated that pacritinib reduced cell growth in AML cell lines and primary samples with different genetic mutations. Furthermore, it has been shown that IRAK1 inhibitor also suppresses the TLR/IL1R/NF-κB/p38MAPK axis, thus eliminating leukemic cells [83,164,165].

Among the kinases that play a key role in promoting tumor proliferation, controlling mitosis progression, there are serine/threonine Aurora kinases, which include Aurora A (AURKA), Aurora B (AURKB) and Aurora C (AURKC) [166]. It is known that Aurora A is overexpressed in AML patients and induces chemoresistance via NF-κB upregulation both in vitro and in vivo [167,168,169,170]. The discovery of Aurora kinases in cancer identified these kinases as a potential therapeutic target in hematological malignancies and Aurora kinase inhibitors of the second generation, including VX-680, are currently in clinical trial for blood cancers. Wade Wei and colleagues demonstrated that Aurora A was essential to induce AML cell survival via TIFA-dependent NF-κB activation [23]. Inhibition of TIFA phosphorylation by Aurora inhibitor VX-680 reduced AML cell growth through downregulation of NF-κB mediated anti-apoptotic genes (i.e., *BCL-2*, *BCL-X_L_*) and inflammatory cytokines and chemokines (i.e., C-X-C Motif Chemokine Ligand 1 (CXCL1), Interleukin-8 (IL-8)), leading to apoptosis of AML cells and increased sensibility to chemotherapy [23,171]. Other Aurora inhibitors (i.e., AZD1152, Danusertib, Alisertib and AMG900) are currently used clinically, but exhibit anti-tumoral activity via NF-κB-independent pathways [172,173,174].

PIM kinases promote NF-κB constitutive activation and at the same time it is regulated by NF-κB. Several PIM kinase inhibitors, such as AZD1208, SMI-4a, SGI-1776, YPC-21440 and YPC-21817, have been shown to inhibit the growth of AML cells in vitro and in vivo [86,88,175,176]. The most advanced PIM1 inhibitor, SEL24/MEN1703, which inhibits both PIM1 and FLT3, is currently in clinical trial for AML patients. The results obtained from the phase I/II study of SEL24/MEN1703 in AML showed a good safety profile and efficacy in relapsed or refractory AML as a single agent, suggesting that this first-in-class dual PIM/FLT3 kinase inhibitor could be a promising new strategy to treat AML patients particularly those with IDH mutations [177,178].

### 10.3. IL-1β Inhibitors

Several cytokines, including Interleukin-1β (IL-1β), are able to stimulate AML cell proliferation via NF-κB activation [179]. Suppression of IL-1β production by resveratrol (trans-3,4′-trihydroxystilbene), a polyphenolic phytoalexin, promotes caspase 3-dependent AML cells apoptosis via inhibition of IL-1β-induced NF-κB activation. The authors demonstrated that resveratrol is also able to downregulate anti-apoptotic proteins such as BCL-X_L_ and BCL-2 as well as to upregulate the expression of the apoptosis inhibitor XIAP. A suppression of tumor cell proliferation was also observed in primary AML cells following treatment with resveratrol [180].

## 11. Inhibitors of NF-κB Interactome

### 11.1. BRD4 Inhibitors

Based on the important role of BRD4 in regulating leukemia gene transcription along with NF-κB, BRD4 specific inhibitors JQ1 (thieno-triazolo-1,4-diazepine) and IBET-151 have been tested for AML treatment. It has been demonstrated that these two compounds promoted apoptosis in primary leukemic cells and reduced tumor growth in different mouse models, likely via NF-κB inhibition and, with regard to IBET-151, also partially via BCL-2 inhibition [18,96,181,182]. In keeping with these observations, Zou and collaborators demonstrated that JQ1 inhibited BRD4/Acetylated p65 interaction and transcription of NF-κB target genes in cancer cells [94]. Studies conducted with other BET inhibitors, such as CPI0610, demonstrated that they reduced NF-κB mediated pro-inflammatory cytokines production in several hematological malignancies [183]. Furthermore, it was shown that the thienotriazolodiazepine compound MK-8628/OTX015 blocked the binding of BRD2/3/4 to acetylated histone H4, promoted G1 cell cycle arrest, and inhibited the MYC and E2F Transcription Factor 1 (E2F1)-dependent gene expression by downregulating the expression of NF-κB, TLR and JAK/STAT (Janus kinase/signal transducers and activators of transcription)-dependent genes such as IRAK1, TLR6, IL-6, Interferon Regulatory Factor 4 (IRF4), and Myeloid differentiation primary response protein 88 (MYD88) in vitro [184]. Likewise, other BET inhibitors (i.e., ABBV-075, FT-110, PLX51107) were tested in preclinical and clinical studies and showed antitumor activity on AML cell lines and patients both as a single agent and in combination regimens, changing the landscape of available targeted therapies [184,185]. However, further studies are needed to understand to what extent their cytotoxic effects are mediated by NF-κB inhibition.

### 11.2. MDM2 Inhibitors

As for BRD4 inhibitors, the molecular mechanism by which MDM2 agents induce cytotoxic effects are still under investigation. Kojima and collaborators demonstrated that nutlin-3 inhibited MDM2 and triggered AML cell and primary cell apoptosis via p53 reactivation and, partially, via NF-κB repression. In particular, the authors showed that this MDM2 antagonist induced cell death in dose-dependent manner in tumor cells, with less toxicity to CD34^+^ progenitors, by inducing p53 dependent transcriptional activation of pro-apoptotic genes and activation of caspases. However, nutlin-3 is also able to induce apoptosis via mitochondrial translocation of p53 and its interaction with BCL-X_L_ [19,186]. Nutlin inhibitors are generally used in combination therapy with chemotherapeutic drugs such as bortezomib and BET inhibitors, in order to increase their cytotoxic activity [19,187].

## 12. Inhibitors of Downstream Effectors of NF-κB

### BCL-2 Inhibitors

It is well-known that NF-κB signaling regulates apoptotic cancer cell death through the activation of the BCL-2 anti-apoptotic gene family, such as *BCL-2*, *Cellular Inhibitor of apoptosis Protein 1/2* (*cIAP1/2*) [8,188,189]. Owing to the key role of this protein family as regulator of apoptosis, indirectly targeting NF-κB by blocking anti-apoptotic effectors could be a potential therapeutic strategy in order to improve patient outcomes, as well as to overcome resistance in relapsed/refractory cancer when used in combination with radiotherapy or chemotherapy [8]. BCL-2 is an oncogene frequently overexpressed in lymphoid and myeloid neoplasms. Overexpression of BCL-2 makes leukemic cells resistant to apoptosis and chemotherapy, and it is associated with worse prognosis [7,8,190]. Venetoclax is a highly selective oral BCL-2 inhibitor that showed anticancer activity in BCL2-dependent leukemia and lymphoma cell lines and mouse xenograft models as well as in primary cells, both as a single agent and in combination [191,192,193,194]. Preclinical studies suggested that venetoclax synergized with hypomethylating agents, such as decitabine or azacytidine in vitro, and destroyed AML LSCs by impacting their energy metabolism [195]. Furthermore, this combination exhibited a promising therapeutic response in high-risk AML patients (i.e., old people, patients with poor cytogenetic risk, and with secondary AML) [196].

## 13. Conclusions

NF-κB plays a key role in the initiation and progression of AML, where it mediates cancer cell survival and chemoresistence by acting in both AML cells and LSCs. Thus, targeting the NF-κB signaling pathway in AML represents an attractive therapeutic strategy. Among the numerous pharmacological approaches targeting NF-κB, there are both natural and synthetic anticancer drugs able to block either NF-κB core elements or upstream regulators/downstream effectors. Although the use of non-specific NF-κB inhibitors has led to significant improvements in the outcome of AML patients, new targeted approaches need to be investigated to reduce side effects that stem from the systemic blockade of NF-κB functions. Therefore, in the near future, it will be important to acquire a better understanding of the specific genetic drivers involved in NF-κB activation in AML, as well as of NF-κB upstream regulators, interactors and downstream effectors. This improved knowledge will permit methodical cataloguing of the full spectrum of NF-κB-driven programs to choose the best targeted option in a cancer-specific manner.

## Figures and Tables

**Figure 1 cancers-14-03557-f001:**
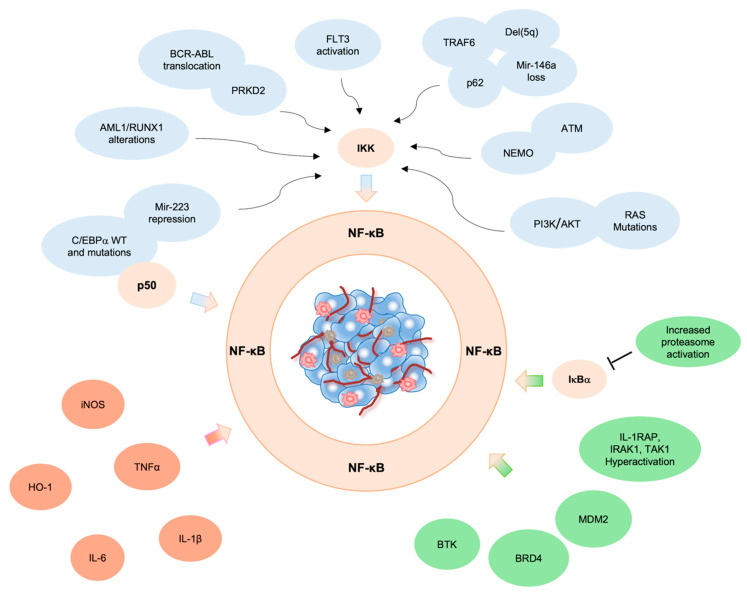
NF-κB activity in AML. Constitutive activation of NF-κB in AML triggered by (***i***) genetic alterations (blue), (***ii***) pro-inflammatory cytokines in the TME (fuchsia) and/or (***iii***) increased expression of NF-κB signaling components (green).

**Figure 2 cancers-14-03557-f002:**
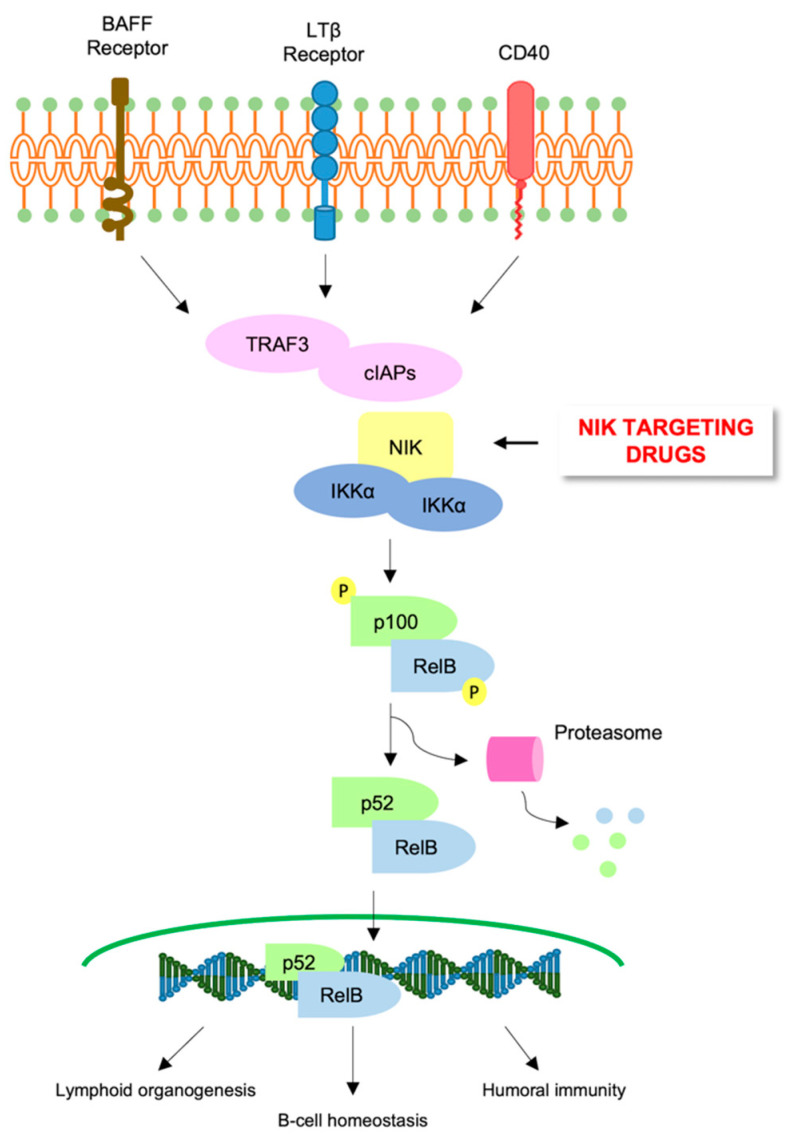
Targeting the NF-κB non-canonical pathway in AML. Schematic representation of the NF-κB non-canonical pathway showing that NIK stabilization suppresses tumor development and could be a potential strategy to treat AML.

**Figure 3 cancers-14-03557-f003:**
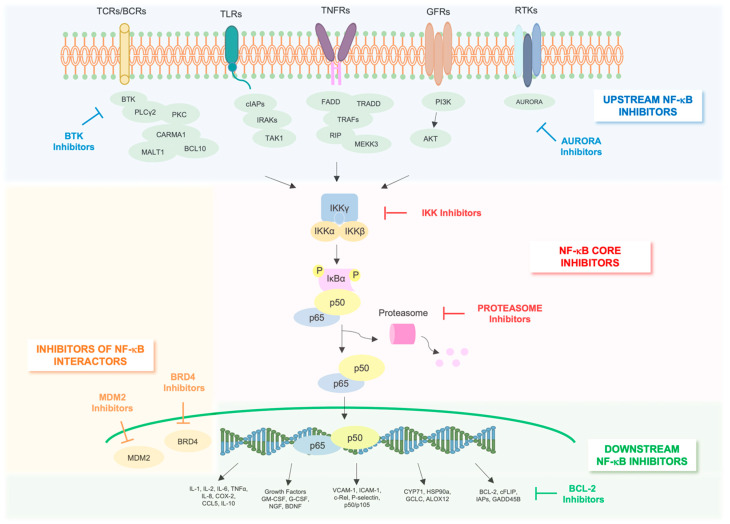
Strategies for therapeutic inhibition of NF-κB canonical pathway in AML. Depicted is the mode of action of major NF-κB inhibitors that are in preclinical development or used in the clinic.

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
