# Peer review of "NF-κB: A Druggable Target in Acute Myeloid Leukemia"

_cancers, 2022, doi:10.3390/cancers14143557_

Round 1

Reviewer 1 Report

In this manuscript, Barbara & Verzella and colleagues reviewed NF-κB as a druggable target in Acute Myeloid Leukemia.

The article is well-written and comprehensive. I have several minor comments:

-          Line 321: I would argue that bortezomib is used routinely in AML, since the current treatment recommendations do not support its use. The citations here do not support the statement either. Perhaps there were some clinical trials worth mentioning, along with their results?

-          PIM kinases are NFkB-inducible oncogenes in AML able to promote survival of AML cells. I would recommend mentioning them in this review as well, since there is currently a quite successful clinical trial of SEL24/MEN1703 PIM inhibitor in AML pts.

-          The role of NFkB in LSCs is the most intriguing from the therapeutic point of view. Could the authors expand this topic more, e.g. in a separate section?

-          The translocation nomenclature should be updated according to the latest recommendations (e.g. BCR::ABL) see: https://www.nature.com/articles/s41375-021-01436-6

-          The subtitle 3.5 is missing a “5” before q

-          Minor English grammar and spelling mistakes should be corrected throughout the manuscript

Reviewer 2 Report

This is a very nice comprehensive review of the NF-KB pathway in AML. My minor edits are listed below.

Line 28 – “while for people >60 years of age the prognosis remains uncertain….” Please correct, it is certainly worse, cure rates <20%

Line 46 – clarify, p65/p50 is part of NK Kapa B ? not listed in the prior sentence

Line 159 – in section 3.5, the subtitle “5q deletion” is misspelled as “q deletion”

Line 192 – FLT3 ITD misspelled

Line 240 – hyperactivation misspelled

Line 451 – tyrosine misspelled

Line 578 – radiotherapy?

Line 588 – 601 : recommend omitting section 12.2 Natural Compounds, essential oils.  No data presented about AML, does not add to this review. 
